# Voltage-Gated T-Type Calcium Channel Modulation by Kinases and Phosphatases: The Old Ones, the New Ones, and the Missing Ones

**DOI:** 10.3390/cells12030461

**Published:** 2023-01-31

**Authors:** Ankush Sharma, Ghazala Rahman, Julia Gorelik, Anamika Bhargava

**Affiliations:** 1Department of Biotechnology, Indian Institute of Technology Hyderabad (IITH), Kandi 502284, Telangana, India; 2National Heart and Lung Institute, Faculty of Medicine, Imperial College London, London W12 0NN, UK

**Keywords:** T-type calcium channels, kinase, phosphatase, modulation, calcium

## Abstract

Calcium (Ca^2+^) can regulate a wide variety of cellular fates, such as proliferation, apoptosis, and autophagy. More importantly, changes in the intracellular Ca^2+^ level can modulate signaling pathways that control a broad range of physiological as well as pathological cellular events, including those important to cellular excitability, cell cycle, gene-transcription, contraction, cancer progression, etc. Not only intracellular Ca^2+^ level but the distribution of Ca^2+^ in the intracellular compartments is also a highly regulated process. For this Ca^2+^ homeostasis, numerous Ca^2+^ chelating, storage, and transport mechanisms are required. There are also specialized proteins that are responsible for buffering and transport of Ca^2+^. T-type Ca^2+^ channels (TTCCs) are one of those specialized proteins which play a key role in the signal transduction of many excitable and non-excitable cell types. TTCCs are low-voltage activated channels that belong to the family of voltage-gated Ca^2+^ channels. Over decades, multiple kinases and phosphatases have been shown to modulate the activity of TTCCs, thus playing an indirect role in maintaining cellular physiology. In this review, we provide information on the kinase and phosphatase modulation of TTCC isoforms Cav3.1, Cav3.2, and Cav3.3, which are mostly described for roles unrelated to cellular excitability. We also describe possible potential modulations that are yet to be explored. For example, both mitogen-activated protein kinase and citron kinase show affinity for different TTCC isoforms; however, the effect of such interaction on TTCC current/kinetics has not been studied yet.

## 1. Introduction

### 1.1. Voltage-Gated Calcium Channels

In the cell, voltage-gated Ca^2+^ channels (VGCCs) play two distinct roles: they can depolarize the membrane potential to promote electrical excitability, and they can also activate transient cytoplasmic Ca^2+^ signals. Action potentials and subthreshold depolarizing impulses cause Ca^2+^ influx through VGCCs, which are found in many different cell types [1]. The Ca^2+^ ions entering the cells can act as secondary messengers, thus accounting for various physiological roles such as ATP synthesis, fertilization, cell cycle, differentiation, proliferation, apoptosis, neurotransmitter/hormone release [2,3,4], activation of gene transcription [5], and muscle contraction [6] thus indicating VGCC roles in non-excitable cells as well. Different cascade effects on cell activities, such as cell proliferation, motility, or even cell death, may result from variations in Ca^2+^ channel transcript levels [7]. Thus, VGCCs play a critical role in controlling the function of several organs, such as the brain, heart, and muscles. Malfunctions of VGCCs can result in a variety of pathophysiological conditions, from cardiovascular diseases to neurological and psychiatric disorders like epilepsy, pain, and autism [8]. Over the last decade or so, VGCCs have been shown to be involved in the disorders of non-excitable cells as well, such as diabetes and cancer [9,10]. VGCCs are classified into two major categories: low-voltage activated (LVA) TTCCs that are activated below the action potential threshold and mainly regulate the excitability of the cells, and high-voltage activated (HVA) channels (L, N, P/Q, and R-types) that are activated after the initiation of an action potential and are therefore important effectors for transient Ca^2+^ signaling [11]. Voltage-gated calcium channels are classified into three subfamilies (Cav1, Cav2, Cav3). Each family has an internal sequence identity of >80% and an interfamily sequence identity of 52% (Cav1 & Cav2) and 28% (Cav3 & Cav1/Cav2), as shown in Figure 1 [12,13,14,15]. TTCCs, being LVA, also have an important role in Ca^2+^ homeostasis through calcium influx near resting membrane potential in several cell types.

### 1.2. T-Type Calcium Channels

TTCCs are LVA channels that belong to the family of VGCCs. There are three isoforms of TTCCs, Cav3.1, Cav3.2, and Cav3.3, that are encoded by the genes CACNA1G, CACNA1H, and CACNA1I, respectively. TTCCs are formed by α1/Cav3 pore-forming subunit, consisting of four homologous domains, each of those made up of six transmembrane helices (S1 to S6). The four domains are linked together with intracellular loops (I-II, II-III, and III-IV) between the S6 segment of the preceding domain and the S1 segment of the following domain. The Ca^2+^ selectivity is due to the acidic residues (glutamate or aspartate) in the pore-forming segments (P loop, connecting segments S5 and S6 of each domain [16,17]. It is known that multiple kinases and phosphatases interact with and thereby modulate TTCCs. These specific kinases and phosphatases are shown in Figure 2 [18,19]. The modulation of TTCCs by these kinases and phosphatases is discussed in detail in the later sections of the article.

TTCCs can operate very close to resting membrane potentials, with Cav3.1 having the most rapid inactivation kinetics (Cav3.1 > Cav3.2 > Cav3.3) [20]. TTCCs have wide expression throughout several organs (Table 1), where they maintain some vital functions such as neuronal excitability, cardiac pacemaking, arterial constriction, hormone release, cell proliferation, cell cycle, etc. [21,22,23,24,25]. In addition, they are also involved in pathological conditions not associated with cellular excitability, such as cancer. Some of the pathological roles of TTCCs are mentioned below in Table 2.

### 1.3. Protein Kinases

Protein kinases (PKs) are molecular switches that exist either in an “on” state with maximum activity or an “off” inactive state. Remarkably, PKs have striking structural similarities in the “on” state while different PKs in the “off” state can adopt distinct conformations [52]. All PKs catalyze the same reaction, the transfer of the γ -phosphate of ATP to the hydroxyl group of serine, threonine, or tyrosine. Eukaryotic PKs (EPKs) constitute one of the largest recognized protein families represented in the human genome [53]. About 1.7% of all human genes, or 518 distinct PKs, are thought to be encoded by the human genome [54]. The catalytic domain of EPK contains 12 preserved subdomains. The invariant aspartate in the subdomain VIB, which is most likely the catalytic base in the phosphotransferase activity, and the invariant lysine in subdomain II, which binds and orients ATP, are indispensable for the catalytic function of EPKs.

In terms of substrate selectivity, PKs may be divided into three major categories: Ser/Thr PKs (PKs that utilize serine/threonine residue as the principal phosphate acceptor site), protein tyrosine kinases (PKs that utilize tyrosine residue as the principal phosphate acceptor site), and dual-specificity PKs (e.g., mitogen-activated protein kinase (MEK) that phosphorylate both threonine and tyrosine residues on target proteins) [55]. PKs are essential for controlling various cellular events such as cell–cell adhesion, cell–substrate adhesion, and hormone signaling that are associated with the dysregulation of kinase-related signaling and may lead to cancer and developmental disorders [56,57]. 

### 1.4. Protein Phosphatases

The action of PKs in the cell is balanced by protein phosphatases (PPs), which remove phosphate groups from the amino acid residues to manage the regulation of protein function. In contrast to PKs, eukaryotes utilize a number of unique molecular paradigms for the construction of the catalytic components in the PPs with which they carry out the dephosphorylation of phosphoproteins. PPs have high evolutionary diversity and complexity based on different/unrelated ancestors compared to PKs [58,59,60]. They also have an intrinsic substrate profile, as phosphatases can recognize proteins, lipids, or carbohydrates. PPs have been divided into protein Ser/Thr phosphatases (PSTPs or PSPs) and protein tyrosine phosphatases (PTPs) based on their unique catalytic processes and substrate selectivity [61,62].

## 2. Kinase Modulators of T-Type Ca^2+^ Channels

### 2.1. Modulation of T-Type Ca^2+^ Channels by Serine/Threonine Kinase Family

#### 2.1.1. Modulation of T-Type Ca^2+^ Channels by Protein Kinase A

Protein kinase A (PKA) is one of the first PKs to be identified [63], sequenced [64], cloned [65], and had crystal structure determined [66]. It is a Ser/Thr kinase that participates in a number of biological processes through phosphorylating and, thus, regulates the activity of many proteins, which also includes the regulation of TTCCs. In mammalian cells (tsA-201-kidney cell line and Chinese hamster ovary cell line), PKA enhanced Ca^2+^ current through all three TTCC isoforms in a temperature-dependent manner. PKA effects were observed at physiological temperatures ranging from 30–37 °C but not at room temperature (22–27 °C). This could be due to the impaired kinase translocation at room temperature [67]. Several studies demonstrated PKA-mediated upregulation of current through TTCCs though very few of them demonstrated isoform-specific or direct modulation of TTCCs. In rodent trigeminal ganglion neurons, activation of both neuromedin B receptor and brain-derived neurotrophic factor receptor activated PKA, which stimulated current through Cav3.2 but not Cav3.1 and Cav3.3 TTCCs to induce hyperexcitability and pain sensitivity. Blockade of TTCC reduced these effects [68,69]. In mice, trigeminal ganglion neuronal cells, G-protein coupled receptor (GPCR) 30 (GPR30) estrogen receptor activation also led to the activation of PKA, which further enhanced current through TTCCs [70], indicating that downstream effects of GPCRs are also mediated by TTCCs in neuronal tissues. Serotonin-induced activation of 5-hydroxytryptamine receptor-7 (5HT_7_) receptors in rat glomerulosa cells enhanced current through TTCCs via a PKA-mediated pathway, which led to aldosterone release [25,71], indicating a role of TTCCs in hormone release in non-excitable cells. Further evidence for the PKA role in hormone release via TTCCs in non-excitable cells comes from human adrenocortical cells where cortisol secretion upon activation of 5HT_4_ receptors was dependent on the activation of cAMP-dependent PKs that enhanced membrane Ca^2+^ influx through TTCCs to activate other VGCCs ultimately leading to the increased intracellular Ca^2+^ and, thus cortisol secretion [72]. A similar observation was made in the primary cultured ovine and human somatotrophs, where growth hormone-releasing hormone (GHRH)-induced T-type current was blocked by a PKA inhibitor. This PKA-mediated influx of Ca^2+^ served as a key step in growth hormone exocytosis [73]. Luteinizing hormone (LH), a major regulator of testosterone synthesis, stimulates Leydig cells, which involves the augmentation of TTCC [74]. The effect of LH is completely blocked upon application of PKA inhibitors H89 and a synthetic inhibitory peptide (IP-20), indicating a PKA-mediated augmentation of T-type current [75]. In an attempt to identify the structural domains that were essential for PKA modulation of TTCCs, chimeric Cav3.2 channel was over-expressed in Xenopus oocytes, which revealed that PKA increases Cav3.2 channel current by a possible modulation in its II-III cytoplasmic loop [76] which was later identified as ser1107. This information is lacking for other Cav3.1 and Cav3.3. Fibroblast growth factor-23 (FGF-23), when bound to the fibroblast growth factor receptor type-1 (FGFR-1) in adult mice, selectively increased TTCC. A PI3K-dependent PKA signaling cascade was discovered through pharmacological manipulation. Because TTCCs are known to play a role in pain transmission, FGFR1 antagonists can be effective analgesics for FGF-23-induced nociception. [77]. An interesting study on NIH 3T3 cells stably expressing five muscarinic receptor subtypes (m1-m5) demonstrated a differential modulation of T-type current upon acetylcholine (Ach) application. Application of Ach on cells expressing m3 and m5 receptors showed an increase in the T-type current. PKA inhibitors blocked this effect, indicating a PKA-mediated augmentation of TTCC upon m3 and m5 receptor activation. While m1 receptor stimulation activated PKA, it did not increase T-type current. An increase in T-type current was observed upon preincubation of cells with PKC inhibitors implying a crosstalk between PKA and PKC. Hence, fine-tuning the activity of these kinases is essential for the modulation of TTCC.s On the other hand, no significant change in the amplitude of T-type current was observed upon m2 and m4 receptor activation [78]. Recently, a study with mesenchymal stem cells (MSCs) transfected with high mobility group box 1 (HMGB1) demonstrated an enhanced T-type current via a PKA-mediated pathway. Here PKA was not shown to interact directly with TTCC, but it modulated TTCC indirectly by phosphorylating β-catenin, which increased its translocation to the nucleus. Phosphorylated β- catenin increased the transcription of γ- cystathionase, which catalyzes H2S production, in turn augmenting TTCC, specifically Cav3.2. This augmentation of Cav3.2 TTCC by HMGB1 increased the mobility of MSCs, which can help in clinical applications in tissue repair and regeneration [79]. Protein expression of Cav3.1 and Cav3.3 protein was not detected in transfected MSCs and therefore ruled out PKA modulation of Cav3.1 and Cav3.3 in MSCs.

PKA modulation of TTCCs does not always enhance current through TTCCs as described above, but a handful of studies have demonstrated PKA-mediated downregulation of current through TTCCs. Dopamine, a neurotransmitter and hormone, inhibited T-type currents in bass retinal horizontal cells, an effect that is prevented by PKA inhibitors and mimicked by 8-CPT, a cAMP analog. PKA modulation of these currents may influence the response properties of horizontal cells [80,81]. A similar observation was made in newt olfactory cells where adrenalin stimulation inhibited T-type current in PKA dependent manner and enhanced odorant contrast in olfactory perception [82]. A subsequent study in H295R cells (human adrenocortical carcinoma cells) revealed that dopamine binding to dopamine receptors 1 and 2 inhibits Cav3.2 channel currents by Gβγ dimers. PKA functions as a molecular switch for the function of Gβγ dimers where inhibition by Gβγ dimers requires PKA-mediated phosphorylation of Cav3.2 at ser1107 residue in the II-III loop of the channel. Failure of this mechanism might be a contributing factor for aldosterone excess and loss of dopaminergic tone in some forms of human hypertension, thus implicating PKA regulation of Cav3.2 as a causative factor. Notably, PKA phosphorylation site ser1107 is specific to Cav3.2 since swapping the II-III loop of Cav3.2 with Cav3.1 eliminated the PKA modulation of the Cav3.2 channel [83]. This is surprising given the fact that ser1107 is conserved across all three isoforms and, thus, warrants further investigation for identifying the phosphorylation sites in Cav3.1 and Cav3.3. In mouse dorsal root ganglion (DRG) neurons, neuromedin U, a neuropeptide with diverse physiological functions, inhibited current through TTCCs in a dose-dependent manner that required PKA activity. This revealed a novel role for neuromedin U peptide in neuronal hypo-excitability in small DRG neurons in mice [84]. PKA-mediated inhibition of current through TTCCs was also observed upon muscarinic M4 receptor stimulation by alpha-cobratoxin (a neurotoxic protein) in DRG neurons [85]. In these neurons, pre-treatment of cells with a PKA inhibitor or selective muscarinic M4 receptor antagonist, tropicamide, abolished the inhibition of TTCC current observed upon alpha-cobratoxin application, suggesting that the analgesic effects of alpha-cobratoxin are mediated by its effect on TTCC current in DRG neurons. Reduction in TTCC current via the action of PKA was observed in rat retinal ganglion cells, which was mediated by activation of G-protein coupled cannabinoid receptors CB2 but not cannabinoid receptors CB1. Cannabinoid receptors are involved in multiple neuronal functions, and these functions are in part regulated by their effect on downstream targets such as TTCCs. This study also points to the fact that the action of PKA on TTCCs is determined by the specific GPCR type and isoform [86]. 

Overall, we observed that PKA showed differential regulation of TTCCs, and enhancement or reduction of current, which seems to be tissue and GPCR-specific. Therefore, a generalization cannot be made on PKA regulation of TTCCs. Reduction in the TTCC current was observed when binding the ligand to a GPCR led to the activation of a G-inhibitory alpha subunit, whereas activation of the G-stimulatory alpha subunit led to the enhancement of current through TTCCs [70,84]. 

#### 2.1.2. Modulation of T-Type Ca^2+^ Channels by Ca^2+^/Calmodulin-Dependent Protein Kinase II

Calmodulin is a 16.7 kDa protein highly conserved in eukaryotes. It was discovered as a Ca^2+^ regulator in the brain. When Ca^2+^ binds to calmodulin, it induces a structural modification, forming a Ca^2+^/calmodulin complex [87]. The Ca^2+^/calmodulin complex activates Ca^2+^/calmodulin-dependent protein kinase II (CaMKII), which participates in a number of signaling cascades, including those that regulate chloride transport [88], immune cell activation [89] and Ca^2+^ homeostasis [90]. It plays a significant role in long-term potentiation [91], memory, and cardiac function [92]. CaMKII dysregulation is also associated with a number of cardiac diseases, such as dilated cardiomyopathy and arrhythmia [93,94]. CaMKII works in a feedback loop where its activity is regulated by a number of Ca^2+^ channels [95], and in turn, it regulates multiple channel proteins [96]. In calf adrenal glomerulosa cells, the elevation of intracellular Ca^2+^ concentration led to the enhancement of TTCC current, which was abolished upon application of a CaMKII inhibitor, suggesting that CaMKII could enhance current through TTCCs. In addition, CaMKII also induced a leftward shift in the activation threshold of TTCCs, rendering them suited to activation by a low amplitude depolarizing stimulus [97]. In fact, this could be how TTCCs can be involved in maintaining calcium influx at near resting membrane potentials. CaMKII regulation could further enhance current through TTCCs in pathological conditions where cells are slightly depolarized. Subsequent experiments in adrenal glomerulosa cells also reiterated that CaMKII could enhance the unitary current amplitude of TTCCs, which was dependent on intracellular Ca^2+^ concentration [98]. CaMKII, however, showed differential regulation of Cav3 family members, augmenting only Cav3.2 but not the Cav3.1 isoform in a heterologous overexpression system using HEK293 cells [99]. This augmentation of TTCC current is predicted to be necessary for sustaining stimulated aldosterone production by glomerulosa cells. Mechanistic studies using the heterologous over-expression system HEK293 revealed that CaMKII could phosphorylate serine residue in the Cav3.2 II-III intracellular loop, which is absent in the Cav3.1 subunit, thus providing a basis for the differential regulation of these channels by CaMKII [100]. This mechanism was later confirmed in vivo in rat adrenal glands where angiotensin II receptor stimulation activated CaMKII, which then phosphorylated ser1198 in Cav3.2 channels leading to the augmentation of Cav3.2 current and sustained aldosterone secretion [101]. The involvement of TTCCs in aldosterone release has great significance in cardiovascular disease, where there is a marked activation of the renin-angiotensin-aldosterone axis. TTCC regulation by the CaMKII pathway is involved in several different functions other than hormone secretion. It was observed that spiro[imidazo [1,2-a] pyridine-3,2-indan]-2(3H)-one (ST101), a cognitive enhancer, augments calcium influx through TTCCs in the brain cortex, thereby causing autophosphorylation of CaMKII leading to long-term potentiation as observed in rat cortical slices but not in hippocampal slices. This is also an example where activation of CaMKII is dependent on TTCC activity, contrary to CaMKII augmentation of TTCC currents. This CaMKII activation was necessary for direct phosphorylation of AMPA-type glutamate receptor subunit 1 (GluR1) in AMPA receptors for long-term potentiation (cognitive enhancement), thus suggesting a role of TTCCs in the cognitive enhancement and a potential therapeutic target in diseases such as Alzheimer’s [102]. SAK3, another spiroimidazopyridine derivative that is also an enhancer of TTCC Cav3.1, showed anti-depressant-like activity, which was in part mediated by CaMKII. SAK3 enhancement of Cav3.1 in olfactory bulbectomized mice led to the rescue of reduction of CaMKII phosphorylation levels that were observed in untreated olfactory bulbectomized mice [103]. This reiterates the involvement of TTCCs in cognitive disorders and suggests that modulation of TTCCs may be targeted for therapeutic purposes. Similar to PKA-mediated reduction in TTCC current upon activation of cannabinoid receptors, CaMKII has also been shown to dampen the activity of TTCCs. In retinal muller cells, CaMKII inhibited currents carried by TTCCs upon activation of cannabinoid receptors, both CB1 and CB2, unlike PKA, which reduced TTCC current only upon activation of CB2 receptors, as mentioned in the previous section. This demonstrated that endogenous cannabinoids could regulate TTCCs by acting on multiple receptors and by employing multiple pathways [86]. The mechanism underlying the off-target neurotoxic effect of vincristine (VCR), a chemotherapeutic drug, was recently investigated using a mouse model, which revealed that VCR-induced neuropathic pain was associated with astrocyte activation. This astrocyte activation was accompanied by an increase in the expression of several genes, including CaMKII, Cav3.2, and connexin-43. CaMKII increased Ca^2+^ influx via the Cav3.2 channel, resulting in connexin-43-dependent inflammation. However, siRNA-mediated knockdown of connexin-43 had no effect on astrocyte activation or the expression of Cav3.2 and CaMKII [104], indicating a rather consequential pathway.

In summary, it is observed that CaMKII maintains Ca^2+^ homeostasis in both excitable and non-excitable cells via a feedback loop and by acting on several protein substrates, one of which is TTCCs. By regulating TTCCs, CaMKII participates in a variety of cell functions such as cognitive enhancement, endocannabinoid signaling, aldosterone secretion, etc. CaMKII, unlike other kinases, targets only Cav3.2 and not Cav3.1, as demonstrated in a heterologous expression system and an in vivo rat model. CaMKII-mediated modulation of TTCCs may serve as a potential therapeutic target in cardiovascular and cognitive disorders due to its involvement in multiple aforementioned pathways. Whether CaMKII can phosphorylate Cav3.3 remains elusive. Also remaining undetermined is the modulation of TTCCs by other CaMKs.

#### 2.1.3. Modulation of T-Type Ca^2+^ Channels by Rho/Rho-Kinase

Rho kinases (ROCK) are recognized as downstream effector proteins of the small GTPase Rho [105]. They primarily participate in cytoskeleton rearrangement and cytoskeleton-associated processes [106,107,108,109,110]. The signaling lipid molecule lysophosphatidic acid (LPA) is an endogenous ligand for a class of GPCRs that mediate downstream effects by activating ROCK. Studies on LPA receptors using LPA receptor knockout mice and ROCK inhibitors suggested that Rho kinase is associated with neuropathic pain [111]. Since TTCCs are also involved in neuropathic pain, the association of TTCCs and ROCK was investigated though data is very limited. Application of LPA in cells overexpressing Cav3.1 and Cav3.3 TTCCs inhibited TTCC current via phosphorylation of amino acid residues in the II-III linker region of the channel that was dependent on ROCK activity as TTCC inhibition was abolished upon using ROCK inhibitors. Cav3.2 TTCC showed a more complex regulation by ROCK where both inhibition and augmentation of Cav3.2 current could be observed with depolarizing shift which awaits further investigation. LPA reduced the native current carried by TTCCs in Y29 retinoblastoma and lateral habenular neurons. This modulation was found to be cell-specific, as in DRG, the current carried by native Cav3.2 was found to be upregulated [112]. 

Further, a study in rat cardiomyocytes showed hypoxia-induced selective upregulation of the Cav3.2 channel and an increased ROCK1 expression. This stabilization of Cav3.2 mRNA and subsequent increase in T-type current is mediated by a hypoxia-induced factor 1α (HIF-1α). Application of C3 transferase, a Rho inhibitor, partially blocked HIF-1α stabilization of Cav3.2 mRNA, whereas HIF-1α mutant did not affect RhoA activity which indicated an upstream regulation of HIF-1α by RhoA. Moreover, C3 transferase and some ROCK inhibitors, Y27632 and fasudil, significantly reduced hypoxia-induced upregulation of Cav3.2 mRNA. These findings suggest that ROCK1 regulates hypoxia-induced upregulation of Cav3.2 current via HIF-1 α, which might be involved in the increased arrhythmias seen in ischemic conditions as well as the pathogenesis of diseases associated with hypoxic Ca^2+^ overload [113]. Potent regulation of TTCCs by ROCK activation suggests that this modulation could be important for neuronal physiology. However, more investigations using in vivo models are needed before we can exploit this modulation for any therapeutics. In addition, ROCK has two isoforms, ROCK1 and ROCK2. It would be interesting to determine if any specific isoform of ROCK is involved in the modulation of TTCCs by ROCK, which needs isoform-specific blockers of ROCK.

#### 2.1.4. Modulation of T-Type Ca^2+^ Channels by Protein Kinase G

Protein kinase G (PKG) is a PK that is activated by cGMP and regulates a number of downstream targets. PKG plays a crucial role in the control of blood pressure. PKG also mediates Ca^2+^ homeostasis [114,115] and modulates platelet function and adhesion [116,117]. It also facilitates the release of renin [118] and causes relaxation of vascular smooth muscle accompanied by a decrease in the TTCC current. PKG-mediated downregulation of TTCC was also observed in rat cerebral arterial smooth muscle upon nitric oxide (NO), a vasodilator, application. NO-mediated suppression of TTCCs via PKG influences Ca^2+^ dynamics, restraining its entry through TTCCs and limiting its contribution to myogenic tone. Hence NO/PKG/TTCC controls smooth muscle function where the loss of NO-induced suppression of TTCC may serve as an underlying mechanism for cerebral vasospasm [119,120]. Data regarding PKG regulation of TTCCs is very limited. In an early study using NG108-15 cells (a neuroblastoma x glioma hybrid cell line), it was observed that cGMP had no effect on TTCC activity [121]. However, further studies using Newt olfactory cells revealed that cGMP increased Ni^2+^-sensitive current carried by TTCCs. This enhancing effect of cGMP was mimicked by the application of a cGMP phosphodiesterase inhibitor or the permeant cGMP analog, CPT-cGMP. Whereas KT5823, a PKG inhibitor, abolished the cGMP-mediated increase in TTCC current, indicating a PKG-mediated increase in the TTCC current. It is suggested that cGMP via augmentation of TTCCs may lower the threshold in olfactory perception and also prevent the saturation of odor signals by increasing the maximum spike frequency [122]. As with other Ser/Thr kinases, PKG also showed dual regulation where, in addition to augmentation as described above, it also mediated inhibition of TTCC current via activation of GPCR sst5 in acutely dissociated retinal ganglion cells [123]. Since TTCCs are important determinants of neuronal excitability, this may provide a mechanistic basis for the protection of ganglion cells against Ca^2+^ overload and, thus, injury. Studies are required for the elucidation of PKG phosphorylation sites in TTCCs and isoform-specific modulation of TTCCs by PKG if any. 

#### 2.1.5. Modulation of T-Type Ca^2+^ Channels by Protein Kinase C

In the heart, activation of various GPCRs is important to maintain cellular excitability and homeostasis. It is thus important to know the downstream effectors of such GPCR activation. The angiotensin pathway is already a target in cardiovascular disorders; therefore, the effect of the angiotensin pathway on TTCC current was investigated. Since Protein kinase C (PKC) is also activated upon activation of angiotensin receptors [124], the crosstalk between receptor activation and TTCC was investigated. PKC is a Ser/Thr kinase that selectively regulates Ca^2+^ and K^+^ currents [125]. Just like PKA, PKC is shown to modulate all three isoforms of TTCCs in a highly temperature-dependent manner [67]. TTCC isoforms are also differentially regulated upon PKC activation. In both human and chick cardiomyocytes, angiotensin II enhanced current through TTCCs, an effect that was mimicked by a PKC activator, phorbol 12,13-dibutyrate [124]. Further studies on cardiomyocytes revealed PKC-dependent elevation of Cav3.2 current but not Cav3.1 current in response to lysophosphatidylcholine-induced stimulation of PKC in a concentration-dependent manner [126]. These studies are important in light of the fact that PKC inhibitors are also being researched as therapeutic drugs in diabetes-related hypertension or endothelial insulin resistance [127], that is, disorders where Ca^2+^ channels are also implicated. Upregulation of current through TTCCs was observed in Leydig cells upon luteinizing hormone induction. Although this effect is mostly mediated by PKA, partial blockage of TTCC current by chelerythrine (PKC inhibitor) indicated a role also for PKC in Leydig cell Ca^2+^ dynamics [75]. PKC-mediated augmentation of TTCC was found to be involved in endothelin 1-induced calcium entry in neonatal rat ventricular myocytes [128]. Endothelin 1 is a potent endogenous vasoconstrictor and might regulate blood pressure and contribute to hypertrophy through this pathway. Studies on Xenopus oocytes revealed that phorbol 12-myristate 13-acetate (PMA), a PKC activator augmented the current amplitude of all three TTCC isoforms but to a varying degree. This augmentation was inhibited by PKC inhibitors. However, a similar effect of PMA was not seen in other mammalian cell lines indicating a species-specific PKC regulation. An increase in the Cav3.1 current by PMA in xenopus oocytes was not brought about by a change in channel surface density. It could be due to direct PKC phosphorylation of Cav3.1 in the intracellular II-III loop region or by indirect consequence of phosphorylation of other signaling effectors [129,130].

Numerous investigations have also demonstrated PKC-mediated suppression or downregulation of TTCCs and currents through them though this was mostly observed in either heterologous overexpression systems or neuronal tissues. Earliest studies on chicken DRG neurons, rat hippocampal neurons, canine cardiac Purkinje and ventricular cells, rat DRG neurons, and rat pituitary GH3 cells showed an inhibitory effect of PKC activators-diacylglycerol (DAG) analog 1-oleoyl-2-acetyl-sn-glycerol (OAG) and PMA on the Ni^2+^ sensitive T-type current [131,132,133,134,135,136,137]. In contrast to cardiomyocytes, angiotensin II inhibited TTCC in bovine adrenal glomerulosa cells, and this effect was mimicked by PKC activators (phorbol esters and DAG), causing a reduced cytoplasmic Ca^2+^ level indicating a PKC-mediated decrease in the intracellular Ca^2+^ and subsequent release of aldosterone in these cells [138]. Hence, PKC can serve as a potential target for hypertension. In an MN9D dopaminergic cell line, the application of corticotropin-releasing factor (CRF) related peptide, urocortin 1, reversibly inhibited current through TTCCs. Patch clamp studies showed that CRF receptor activation and subsequent downregulation of T-type current were blocked by the application of PKC inhibitor chelerythrine and enhanced by PMA [139]. CRF receptors are predominantly expressed in the dopaminergic neurons, where TTCCs are also one of the important determinants of Ca^2+^ homeostasis and physiology. There is evidence from a rat Parkinson’s model that TTCCs are involved in the pathology of Parkinson’s disease since the application of TTCC blockers showed reduced motor deficits in the Parkinson’s rats. [140]. Therefore, this PKC-mediated suppression of TTCCs via CRF receptor activation could be of significance in diseases such as Parkinson’s, where dopaminergic neurons are affected. In heterologous over-expression systems, PKC is also shown to inhibit TTCC current mainly through inhibition of Cav3.2 channel current upon activation of neurokinin 1 receptor [141]. In an interesting study using HEK293 cells and DRG neurons, ethanol was shown to reduce the current density of Cav3.2 TTCC isoform and produce a hyperpolarizing shift in the steady-state inactivation of the channels leading to an increase in the window current. Since DRG neurons are known to be enriched in Cav3.2, it is speculated that results from HEK293 cells corroborated well with the native channels. The inhibitory effect of ethanol was blocked by myristoylated PKC peptide inhibitor (MPI), whereas a lower concentration of PMA mimicked this effect, suggesting the PKC-dependent activity of ethanol [142]. In contrast, studies with rat cardiomyocytes showed a PKC-mediated enhancement of the TTCC activity upon ethanol administration. A mechanistic study revealed PKC-mediated hyperphosphorylation of GSK3β, which further activated calcineurin-NFAT-Csx/Nkx2.5 signaling. This may serve as a potential mechanism for ‘holiday heart syndrome’ caused by excessive alcohol consumption [143]. In a recent study, the importance of regulation of TTCCs by a specific isoform of PKC was highlighted, where melatonin (a hormone secreted by the pineal gland and important for circadian rhythm) inhibited Cav3.2 TTCC activity through the melatonin receptor 2 coupled to PKC-eta (PKCη) signaling. This downregulation of Cav3.2 decreased the membrane excitability of trigeminal ganglion neurons and pain hypersensitivity in mice, thus providing a mechanistic basis for the protective effects of melatonin. Melatonin did not alter currents through other TTCC isoforms indicating that Cav3.2 is the main target for PKCη [144]. Various studies have suggested crosstalk between PKC and PKA wherein one might inhibit the effects of the other, as seen in T-type current modulation by acetylcholine via muscarinic receptor-1 activation [78]. However, a synergistic effect of both kinases may also be involved in regulating physiological functions [81].

Taken together, it appears that PKC modulation of TTCCs is best suited to the cells or tissues in which they express and based on their contribution to cell physiology. Inhibition of TTCC currents appears to be a protective mechanism employed mainly by neurons to avoid hyperexcitability-related cell death. Upregulation of TTCCs by PKC in cells such as cardiovascular cells appear to be pathogenic and provides a mechanistic basis for the use of GPCR blockers for cardiovascular disorders. 

#### 2.1.6. Modulation of T-Type Ca^2+^ Channels by Cyclin-Dependent Kinase 5

Cyclin-dependent kinase 5 (Cdk5) is a proline-directed Ser/Thr PK meaning this Ser/Thr kinase activity is directed by an adjacent proline, exhibiting a recognition sequence of -X- Ser/Thr-Pro-X-. Based on its 60% sequence similarity and comparable substrate specificities to Cdc2 (a regulator of cell cycle progression), Cdk5 was first assumed to have a role in cell cycle control [145,146,147]. Cdk5 is a neuron-specific kinase and has been most extensively studied for its roles in neuronal migration, neurite outgrowth, axonal guidance, and synaptic plasticity; however, recently, it has been discovered that Cdk5 also plays a major role in circadian clock regulation under physiological conditions, DNA damage, cell cycle re-entry, mitochondrial dysfunction, and oxidative stress [148,149,150,151].

The modulation of TTCCs by Cdk5 was recently shown, where currents through TTCCs were found to be markedly increased by overexpression of Cdk5 in N1E-115 cells (neuroblastoma cell line), while Cdk5 siRNA knockdown significantly decreased TTCC currents. Similarly, overexpression of Cdk5 upregulated Cav3.1 and Cav3.2 TTCC currents in HEK-293 cells expressing Cav3.1 or Cav3.2, respectively [152]. Mechanistically, a phosphorylation site serine 2234 was identified in the c-terminus of Cav3.1 as being phosphorylated by cdk5, whereas the target sites in Cav3.2 were residues serine 561 and serine 1987 [153]. These results suggest that cdk5 may be used to target neuron-specific cellular effects mediated by TTCCs. This warrants further investigation of cell physiology assays modulated by cdk5 and TTCCs together. 

### 2.2. Modulation of T-Type Ca^2+^ Channels by Protein Tyrosine Kinase Family 

Protein tyrosine kinase (PTK) catalyzes the transfer of phosphate from ATP to tyrosine residues on the protein substrates [154]. PTK modulation of TTCCs has been reported in different cell types, which was deduced using PTK blockers. One of the first reports of TTCC modulation by PTK demonstrated that, in mouse spermatogenic cells, TTCC current is enhanced by PTK inhibitors tyrphostin A47 and A25, whereas tyrosine phosphatase inhibitors phenyl arsine oxide and sodium orthovanadate inhibited TTCC currents. In spermatogenic cells, TTCCs are maintained at a low conductance state by tyrosine phosphorylation, and phosphatase activity enhances the conduction, which may serve as a plausible mechanism for sperm activity during the early stage of mammalian fertilization [155]. In another study on NG108-15 cells, genistein, along with two other PTK inhibitors, lavendustin A and herbimycin, suppressed current through TTCCs, which was independent of G-protein activation [156]. This study, however, lacks information on the direct effects of PTK inhibitors on TTCC currents since PTK inhibitors can inhibit and alter the kinetics of TTCCs directly and independently of PTKs [157,158]. There is much need for data regarding PTK modulation of TTCCs. Perhaps, better inhibitors are required which do not block the TTCCs directly. 

## 3. Modulation of T-Type Ca^2+^ Channels by Phosphatases 

Protein phosphatases are enzymes that remove phosphate groups from proteins, and together with protein kinases, they modulate the activities of the proteins in a cell, thus indirectly contributing to cellular functions. Several phosphatases also regulate the activity of TTCCs, as described below.

### 3.1. Modulation of T-Type Ca^2+^ Channels by Tyrosine Phosphatases

Protein tyrosine phosphatases (PTPs) are a broad class of enzymes found in a wide range of cell types [159]. PTPs are encoded by the largest family of phosphatase genes. The cysteine residue serves as a nucleophile and is required for catalysis in these enzymes, which are identified by the active-site signature motif HCX_5_R [62]. By opposing the actions of PTKs, PTPs play a significant role in signal transduction systems involving tyrosine phosphorylation. Usually, PTP activities are measured in conjunction with PTK activities where antagonistic results are observed. In a similar study involving TTCCs, where PTK inhibitors (Tyrphostin A47 and A25) enhanced Ni^2+^-sensitive T-type current in mouse spermatogenic cells, the use of PTP activity antagonists (phenylarsine oxide and sodium orthovanadate) resulted in a blockade of voltage-dependent facilitation of the TTCC current [155]. This data indicates that PTPs, when activated, can enhance TTCCs in mouse spermatogenic cells. Unfortunately, not all studies which have investigated the modulation of TTCCs by PTKs have further explored the role of PTPs, thus amounting to limited data available.

### 3.2. Modulation of T-Type Ca^2+^ Channels by Calcineurin

Calcineurin (CaN) is a Ser/Thr phosphatase that is activated by increased intracellular Ca^2+^ concentrations [160,161]. CaN play a crucial part in several signaling cascades and directly connects Ca^2+^ signaling to protein phosphorylation states [162]. Structurally CaN is a heterodimer expressed as three different isoforms: αCaN, βCaN, and γCaN. While γCaN is shown to be present in the testis and brain, αCaN and βCaN isoforms are reported to be ubiquitously expressed. In terms of three-dimensional structure, these isoforms are generally made up of two subunits: calcineurin A, which contains the catalytic site and an autoinhibitory domain that is displaced following Ca^2+^ and calmodulin binding, and calcineurin B, which is a Ca^2+^-binding protein [163,164,165,166]. Some of the substrates of CaN are transcription factors, receptors, channels, proteins linked to mitochondria, and proteins linked to microtubules [163]. The interaction between CaN and Cav3.2 was discovered via anti-tag immunoprecipitation. It was found that Calmodulin and 0.1 mM Ca^2+^ enhanced the physical interaction between Cav3.2 and calcineurin, but 2 mM EGTA entirely eliminated it, suggesting that the interaction between Cav3.2 and calcineurin is Ca^2+^/calmodulin dependent. CaN-binding deficient Cav3.2 TTCCs (Cav3.2-9A and Cav3.2-ΔA, made by deleting the region necessary for CaN binding) conducted larger current in Cav3.2 transfected HEK293 cells [167], providing evidence that calcineurin might have a negative modulatory effect of Cav3.2 channel. CaN is involved in the G1-S transition either by initiating a pathway that leads to the accumulation of cyclin D1 [168] or by dephosphorylating the nuclear factor of activated T cells (NFAT), resulting in its translocation to the nucleus [169]. Cav3.2 was also found to be necessary for the G1-S transition (Table 2). It is intriguing to note that both CaN and TTCC have a role in G1-S transition, yet CaN modulates Cav3.2 negatively. This warrants further investigations.

## 4. Kinase and Phosphatase Regulation of TTCCs That Is Plausible but Yet to Be Investigated

With the development of sophisticated techniques and advancement in science, there is evidence to suggest several possible kinase modulations of TTCCs by the latest identified kinases; however, experimental data lacks support. We provide such evidences below and have classified them based on the specific isoform of TTCC which may be involved. Experimental investigations in support of the below evidence are required. We also briefly provide information on the kinases and their functions.

### 4.1. Possible Direct Modulation of Cav3.2 by Mitogen-Activated Protein Kinases

Mitogen-activated protein kinases (MAPKs) are Ser/Thr PKs that regulate evolutionarily conserved signal transduction pathways linked to cellular activities such as proliferation, migration, apoptosis, and differentiation [170]. Abnormal MAPK signaling is extensively implicated in the onset and progression of certain forms of cancer [171]. MAPK15 is the latest identified member of the MAPK family [172] that has recently emerged as a key modulator of autophagy [173], reactive oxygen species generation [174], etc. In fibroblasts that expressed oncogenically active Ras and gain-of-function mutant MEK (MAPK/extracellular signal-regulated kinase (ERK) kinase, which is a direct activator of MAPK), T-type currents were significantly suppressed, and treatment with a MEK-specific inhibitor (PD98059) restored TTCC activity [175]. In a similar study, it was observed that Zinc transporter-1, a putative zinc transporter increases Cav3.1 and Cav3.2 activity by activating Ras-ERK signaling. Endothelin-1, a potent stimulator of Ras-ERK signaling, also increased the surface expression of Cav3.1 channels in murine cardiomyocytes (HL-1 cells), confirmed by surface biotinylation. MAPK cascade inhibitor PD98059 abolished the effect suggesting the involvement of MAPKs [176]. In both these studies, however, the direct effect of MAPK on TTCC was not revealed, although affinity capture-mass spectroscopy has revealed a physical interaction between MAPK15 and TTCC Cav3.2 [177]. The modulation of TTCCs by MAPK, if any, would be interesting to understand as both the proteins have overlapping roles in cancer and vital cellular signaling.

### 4.2. Possible Modulation of Cav3.2 by Never in Mitosis A—Related Kinase

Never in Mitosis A (NIMA)-related kinases or Nrk or NEK are a large group of homologous kinases with related functions and structures. They participate in cellular processes such as cell cycle, cell division, cilia formation, and the DNA damage response, and any disturbance or abnormal expression of NEKs can cause different disorders such as cancer, diabetes, ciliopathies, central nervous system disorders, and inflammation-related diseases [178]. The activity and levels of NEK family proteins vary during different phases of the cell cycle, with their activity being tightly regulated both transcriptionally and post-translationally by multi-stage phosphorylation and proteolysis, and by these mechanisms, NEK activity is largely restricted to a brief window at the G2–M transition. During the S-phase, NEK proteins are inactive, they become activated by phosphorylation during the G2 phase, and the kinase activity becomes highest during the M-phase when they are hyperphosphorylated. The level of NEK proteins is kept in check by the proteolytic destruction during the end of the M-phase [179,180]. Human NEKs contain 11 isoforms from NEK1 to NEK11.

Affinity capture-mass spectroscopy data again revealed Cav3.2 TTCC & NEK4 as an interacting unit [181]. However, the significance of this association has never been explored in live cells, if any. The significance of this interaction lies in the fact that studies show both Cav3.2 and NEK4 to have a role in apoptosis and cell cycle progression [44,182] Whether they both are communicating with each other to execute such cellular functions is not known and needs to be further experimentally explored.

### 4.3. Possible Citron Kinase Modulation of Cav3.3

Historically, most of the isoform-specific studies on TTCCs have focused on two other isoforms of TTCCs, which are Cav3.1 and Cav3.2. Very few studies have investigated Cav3.3 isoform-specific regulation. The exact reasons for this disparity are not clear. However, we speculate that the less widespread expression of Cav3.3 and the late discovery of its roles in cellular functions may have been contributory factors. Interestingly, affinity capture-mass spectroscopy revealed a physical interaction between Citron kinase (CIT-K) and Cav3.3 [183]. CIT-K is a rho-interacting Ser/Thr kinase that functions during cytokinesis, where it promotes the actomyosin contractile ring’s constriction [184]. Recently, a mutation in CIT-K has been shown to induce primary microcephaly [185], emphasizing its importance for brain function. Since Cav3.3 channels have a predominant neuronal expression and microcephaly shows symptoms overlapping with symptoms during the dysfunction of TTCCs, it would be really interesting to determine if CIT-K modulates brain Cav3.3 channels.

### 4.4. Possible Modulation of Cav3.2 by SRC Kinase

Src is the prototype member of non-receptor PTKs whose activity is upregulated in multiple human malignancies. The role of Src is not only associated with maintaining cellular physiology but also in synaptic transmission and remote fear memory in the forebrain [186]. Many studies have shown that activation of kinases of this family mediates Ca^2+^ release, and in turn, their activation is regulated by Ca^2+^ signal via calmodulin [187], similar to CaMKII.

Studies regarding the regulation of TTCCs by Src kinase are minimal, and only indirect evidence exists. Gasotransmitter hydrogen sulphide (H_2_S) was shown to stimulate the growth of neurons by the involvement of TTCCs, Cav3.2 in particular, and Src kinase in NG108-15 cells. This neuritogenic effect of H_2_S was completely blocked by the application of BAPTA/AM(1,2-Bis(2-aminophenoxy) ethane-N, N, N′, N′-tetra acetic acid tetrakis (acetoxymethyl ester), an intracellular Ca^2+^ chelator, or by selective Cav3.2 inhibitors, such as zinc chloride and L-ascorbate [188]. Src kinase phosphorylation was also observed upon application of H_2_S, which was prevented by a Ca^2+^ chelator and Src kinase inhibitor PP2(4-amino-5-(4-chlorophenyl)-7-(t-butyl) pyrazolo [3,4-d] pyrimidine) [189]. Overall, this indicated a role for Ca^2+^_,_ Src kinase, and TTCCs in neuritogenesis; however, the underlying pathway remains elusive. Whether Src phosphorylates TTCCs or whether Ca^2+^ influx through TTCCs is responsible for the activation of Src needs further investigation.

### 4.5. Possible Modulation of Cav3.1 by Phosphatase and Tensin Homolog Deleted on Chromosome 10

Phosphatase and Tensin Homolog deleted on Chromosome 10 (PTEN) is a dual phosphatase with both protein and lipid phosphatase activities [190]. PTEN controls the signals that govern metabolic processes, including glycolysis, gluconeogenesis, glycogen production, lipid metabolism, and mitochondrial metabolism [191]. A significant majority of solid tumors show PTEN mutations, and PTEN is also involved in modeling the tumor immune microenvironment and modulating the DNA damage response [192]. Recently, Cav3.1 mRNA regulation in melanoma cells was shown to be associated with PTEN expression, where Cav3.1 mRNA levels were higher in the case of PTEN-compromised vemurafenib-resistant (Vem-R) melanoma cells, and Cav3.1 mRNA levels were lower in Vem-R cells with PTEN overexpression. This data indicated a negative correlation between PTEN and Cav3.1 expression, which may suggest the downregulation of TTCC expression in Vem-R cells by PTEN [193]. Both PTEN and Cav3.1 have been suggested as protooncogenes, and therefore, it will be interesting to determine if one can regulate the expression of the other, which needs further investigation.

## 5. Conclusions

Ca^2+^ can regulate cellular fates such as proliferation, apoptosis, and autophagy linked to cell cycle and even pathologies like cancer and neuronal disorders. Several cellular proteins, including receptors, pumps, and Ca^2+^ sensing proteins, work together to maintain Ca^2+^ concentrations inside the cell and cellular compartments. TTCCs specialized membrane proteins that play a crucial role in the signal transduction of many excitable and non-excitable cell types. Scientific literature shows that the activity of TTCCs is regulated by several kinases and phosphatases, leading to the enhancement or inhibition of TTCC currents. A general scheme of TTCC modulation by kinases and phosphatases is shown in Figure 3. A closer look reveals that multiple kinases acting on the same region (not the same amino acids) showed different effects on TTCCs [19,194]. These varied observations point towards several facts that (1) TTCCs may have different phosphorylation sites/regions specific for specific kinases. There is a lot of scope for finding such phosphorylation sites. Most of the studies did not reveal the actual binding site of the kinase or phosphatase to the TTCCs. (2) The modulation seems to be dependent on the tissue and cell type, and the fact that cells may have differential expression of TTCC isoforms complicates the issue. Development of isoform-specific blockers, or activators of TTCCs may address the issue but is equally challenging. Knock-down strategies may be employed to determine the isoform-specific role of TTCCs in modulation by kinases and phosphatases. (3) The modulation of TTCCs by kinases or phosphatases may be altered in pathological conditions. It would be pertinent to determine the level of expression of kinases and phosphatases in the cells/tissues under investigation to normalize the effects across varied cell types. In addition, in pathological conditions, so far, much of the focus has been given to cancerous conditions. Therefore, other pathological states, such as diabetes etc., could be explored where TTCCs are also implicated.

In conclusion, there is still an opportunity to explore different regions of TTCCs for their modulation by biological molecules. With the advent of techniques like affinity capture-mass spectroscopy and anti-tag immunoprecipitation, researchers have revealed some new binding interactions as well. However, the functional significance of such interactions is still to be explored. TTCC modulation by kinases and phosphatases could be important in pathology, where the modulation can lead to altered Ca^2+^ influx and thus contribute to pathology. An example of this kind exists for L-type Ca^2+^ channels where phosphorylation of L-type Ca^2+^ channels by CaMKII during heart failure contributes to the pathology, and inhibiting CaMKII is protective [195]. Much remains to be explored for TTCCs during pathological conditions.

## Figures and Tables

**Figure 1 cells-12-00461-f001:**
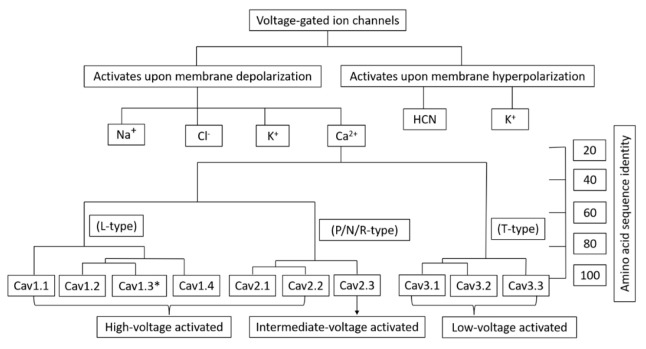
Classification of voltage-gated ion channels based on the activation potential. *Cav1.3 channel isoforms are the exception in L-type channels as they can also be activated at low-voltages.

**Figure 2 cells-12-00461-f002:**
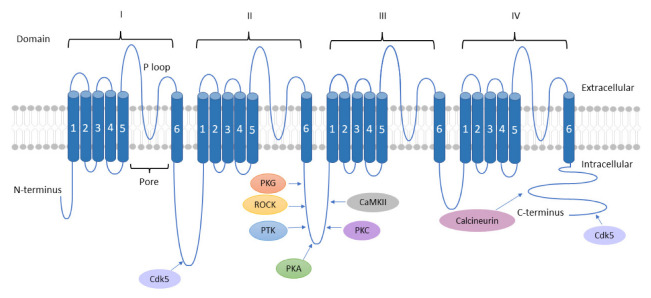
Schematic representation of Cav3 pore-forming subunit showing interaction sites of different kinases and phosphatases. Each of the four domains have transmembrane segments labelled from 1-6. Target residues of PTK, PKG, ROCK, and PKC are not yet identified; however, they are found to be associated with the II-III loop region of the Cav3 channel. PKA phosphorylates ser1107 residue of Cav3.2, while its interaction with other isoforms is not yet known. Ser 1198 of Cav3.2 acts as a modulatory site for CaMKII. Cdk5 interacts with both Cav3.1 (ser 2234 in the C-terminus) and Cav3.2 (ser 561 (I-II loop) & ser 1987 in the C-terminus). Calcineurin recognizes PCISVE (2190–2195) & LTVP (2261–2264) motifs in the C-terminus.

**Figure 3 cells-12-00461-f003:**
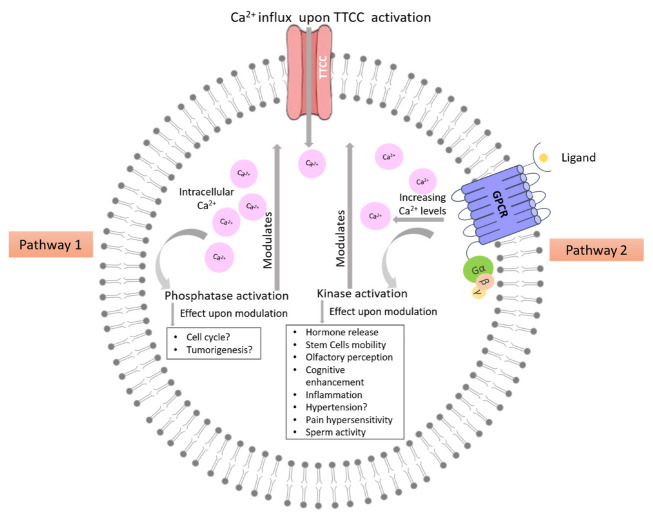
General scheme of TTCC modulation by Kinases and phosphatases. Pathway 1 shows the phosphatase activation via increase in intracellular Ca^2+^ levels, which then modulates the TTCC activity. Pathway 2 shows the kinase activation via GPCR, which in turn modulates TTCC activity. Question mark indicates that direct experimental evidence is awaited.

**Table 1 cells-12-00461-t001:** Major location of T-type Ca^2+^ channels.

T-Type Ca^2+^ Channel Isoform	Location	Reference
Cav3.1	Brain and Peripheral nervous system	[26]
	Ovary and Placenta	[27]
	Heart	[28]
	Pancreatic islets and Beta cells	[29,30]
	Adrenal medulla	[31]
Cav3.2	Brain and Peripheral nervous system	[26,32]
	Heart	[28,33]
	Pancreatic islets and Beta cells	[30,34]
	Kidney and liver	[35]
	Adrenal cortex	[36]
Cav3.3	Brain and Peripheral nervous system	[26,37]
	Pancreatic islets	[30,34]
	Cardiac Purkinje fibers	[38]

**Table 2 cells-12-00461-t002:** Pathological roles of T-type Ca2+ channels.

T-Type Ca^2+^ Channel Isoform	Roles	Reference
Cav3.1 (Encoded by CACNA1G)	CACNA1G as a tumor suppressor gene	[39,40]
	Cav3.1 gain-of-function mutations are associated with childhood-onset cerebellar atrophy.	[41]
Cav3.1, Cav3.2 & Cav3.3	Cav3.1, Cav3.2 & Cav3.3 are involved in the cancer cell proliferation across several cancer types, such as oral squamous cell carcinoma, melanoma, ovarian cancer, colorectal cancer, glioblastoma, neuroendocrine tumors, and esophageal cancer cells as shown by the pharmacological blockade.	[10,42,43,44,45,46,47,48]
Cav3.2 (Encoded by CACNA1H)	CACNA1H mutations are associated with amyotrophic lateral sclerosis and Primary Aldosteronism	[49,50]
	Ca_v_3.2 may be a potential differential biomarker for survival and treatment response in specific breast cancer subtypes	[51]

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
