# Peer review of "Voltage-Gated T-Type Calcium Channel Modulation by Kinases and Phosphatases: The Old Ones, the New Ones, and the Missing Ones"

_cells, 2023, doi:10.3390/cells12030461_

Round 1

Reviewer 1 Report

The review summarizes the current understanding and literature about a single class of voltage gated ion channels and by which protein kinases and phosphatases these are modified. The review serves it purpose in that most studies concerning this topic are comprehensively and systematically presented. I clearly recommend publication of the manuscript, since it will serve as a reference for every newcomer or researcher with an interest to this rather narrow and specialized field. 

Before publication, the manuscript should be revised to improve the accessibility to researchers not familiar with this class of type of ion channels. The manuscript starts with an introduction of this type of channels. Fig.1 presents the schematic structure of Cav3 channels. It contains very little information, because it is not linked to the following part of the manuscript. Phospho-sites or other interesting sites of regulation are not indicated, possibly because they have largely not been mapped, yet. As an outsider of the ion channel field, I would find it helpful, if the Cav3 channels would be introduced by putting them into the systematics of ion channels, at least in comparison to other voltage gated channels concerning their structure and phosphor-regulation. In my view the introduction should better describe the functions of the Cav3 channels. Currently the authors try to summarize this in the form of a table (Tab. 1), organized by single literature references. I find this format not very accessble and legible. The authors should try to focus on common features and principles instead of every single details of the respective reference. 

The authors go on to introduce protein kinases, including a full figure for the general domain organisation of protein kinases. The chaper 1.3 and figure 2 are not informative and useful, because it is disconnected from Cav3 channels and not really needed for a review of Cav3 channels and their phosphor-regulation. 

The central part of the manuscript then systematically lists what is known about a link of Cav3 channels and a series of proteins kinases. These chapters are informative and will serve as a reference. What I miss is a chapter in which common principles or patterns or processes are summarized for the different kinases. 

In the last part the authors speculate on whether a number of additional protein kinases have Cav3 channels as substrates. These paragraphs are in principle fine with me, though they may be more concise, as indirect and speculative interactions are presented. 

Author Response

Reviewer 1

Opinion 1. The review summarizes the current understanding and literature about a single class of voltage gated ion channels and by which protein kinases and phosphatases these are modified. The review serves it purpose in that most studies concerning this topic are comprehensively and systematically presented. I clearly recommend publication of the manuscript, since it will serve as a reference for every newcomer or researcher with an interest to this rather narrow and specialized field. 

Response: We thank the reviewer for the time and appreciating our work.

Opinion 2a. Before publication, the manuscript should be revised to improve the accessibility to researchers not familiar with this class of type of ion channels. The manuscript starts with an introduction of this type of channels. Fig.1 presents the schematic structure of Cav3 channels. It contains very little information, because it is not linked to the following part of the manuscript. Phospho-sites or other interesting sites of regulation are not indicated, possibly because they have largely not been mapped, yet.

Response: We agree with the reviewer and have now modified the old Figure 1 (Figure 2 in the revised manuscript) to contain sites on which different kinases and phosphatases act to modulate TTCCs. We hope that the figure is now more informative.

Opinion 2b. As an outsider of the ion channel field, I would find it helpful, if the Cav3 channels would be introduced by putting them into the systematics of ion channels, at least in comparison to other voltage gated channels concerning their structure and phosphor-regulation.

Response: We thank the reviewer for this comment. To address this, we have now included new Figure 1 in the manuscript which contains a hierarchy of voltage-gated ion channels. They are classified based on their activation potential. In addition, voltage-gated calcium channel classification also depicts primary structural similarity between channels. We hope that the figure is informative for novices.

Opinion 2c. In my view the introduction should better describe the functions of the Cav3 channels. Currently the authors try to summarize this in the form of a table (Tab. 1), organized by single literature references. I find this format not very accessible and legible. The authors should try to focus on common features and principles instead of every single details of the respective reference. 

Response: Taking into reviewer’s suggestion we have added few more sentences describing Cav3 channel functions in the introduction. We have also added new Table 1 to include general information about the location of Cav3 channels. In addition, we have considerably refined old table 1 (Table 2 in the revised manuscript) to generalize the pathological roles of Cav3 channels instead of giving details. Hope this is agreeable to the reviewer. Please see Pages 3 and 4.

Opinion 3. The authors go on to introduce protein kinases, including a full figure for the general domain organisation of protein kinases. The chapter 1.3 and figure 2 are not informative and useful, because it is disconnected from Cav3 channels and not really needed for a review of Cav3 channels and their phosphor-regulation. 

Response: We agree with the reviewer that figure 2 was not very informative and useful, therefore we have now excluded it from the review. However, we still think that section 1.3 gives the reader a good overview of the proteins that carry out the phosphorylation as the review is about phosphorylation of Cav3 channels. The organization of our review includes brief introduction of VGCCs, Cav3, Kinases and phosphatases and therefore eliminating just the kinase introduction would leave a gap. However, taking into reviewer’s suggestion, we have further reduced section 1.3 to make it more concise. Hope this is agreeable to the reviewer. Please see Page 4.

Opinion 4. The central part of the manuscript then systematically lists what is known about a link of Cav3 channels and a series of proteins kinases. These chapters are informative and will serve as a reference. What I miss is a chapter in which common principles or patterns or processes are summarized for the different kinases. 

Response: We thank the reviewer for this comment. Indeed, this was a good suggestion. We have now included Figure 3 that summarizes the general scheme associated with the activation of kinases/phosphatases and their modulation of Cav3 channels.

Opinion 5. In the last part the authors speculate on whether a number of additional protein kinases have Cav3 channels as substrates. These paragraphs are in principle fine with me, though they may be more concise, as indirect and speculative interactions are presented. 

Response: Taking into reviewer’s suggestion, we have further reduced section 4 where speculations are presented. Hope this is agreeable to the reviewer. Please see Pages 12-14.

Overall, we have tried our best to address reviewers’ comments and we hope that the reviewers and editors now find our article suitable for publication. 

Reviewer 2 Report

The manuscript "Voltage gated T-type calcium channel modulation by kinases and phosphatases: the old ones, the new ones and the missing ones." is a review about the existing data on the modulation of T-type calcium channels by different proteins that "switch" the channel between the phosphorylated and dephosphorylated state, better known as kinases and phosphatases. The collected data is well presented and the introduction about T-type calcium channels is well structured. The section that refers to the part of the title: "...the missing ones...", i.e., "4. Kinase and phosphatase regulation of TTCCs that is plausible but yet to be investigated" is perhaps the weaker part of the review in terms of that some references are one-time observations or data from meta-analysis that might not survive further molecular-mechanistic studies. For instance: in line 577: "Interestingly, Affinity Capture-Mass Spectroscopy data revealed CIT-K and Cav3.3 as an interacting unit [166]." This study found that the gene CACNA1I (that codes for Cav3.3 channel) is one in several hundreds of genes that might interact with CIT-K but there are no further experiments to test this possibility.

Figs. 1 and 2 are too basic that they do not add anything to the review. It would be more informative and illustrative to show in Fig. 1, for instance, the potential residues/regions of T-type protein(s) that are targets for the kinases or phosphatases that are mentioned on the main text.

Also Fig. 2 could be improved in some way.

Minor points:

1. Line 38: "...thus indicating VGCC roles in non-excitable cells as well." The only non-canonical function for excitable cells that is listed is "activation of gene transcription". I suggest listing more non-excitable related functions for Ca2+ ions. 

2. The sentence starting in line 386:  "CRF receptors are predominantly expressed in dopaminergic neurons where TTCCs are also important in the physiology of dopaminergic neurons and therefore this PKC mediated suppression of TTCCs could be of significance in diseases such as Parkinson’s where Ca2+ homeostasis is affected." The fact that Ca2+ homeostasis is affected does not imply that T-type calcium channels might be involve. There are also intracellular calcium channels (IP3 and Ryanodine, at least) that could contribute to the calcium homeostasis. The phrase is too speculative.

3. Line 568: I think authors wanted to say: Cav3.1 and Cav3.2.

Author Response

Reviewer 2

Opinion 1. The manuscript "Voltage gated T-type calcium channel modulation by kinases and phosphatases: the old ones, the new ones and the missing ones." is a review about the existing data on the modulation of T-type calcium channels by different proteins that "switch" the channel between the phosphorylated and dephosphorylated state, better known as kinases and phosphatases. The collected data is well presented and the introduction about T-type calcium channels is well structured.

Response: We thank the reviewer for the time and appreciating our work.

Opinion 2. The section that refers to the part of the title: "...the missing ones...", i.e., "4. Kinase and phosphatase regulation of TTCCs that is plausible but yet to be investigated" is perhaps the weaker part of the review in terms of that some references are one-time observations or data from meta-analysis that might not survive further molecular-mechanistic studies. For instance: in line 577: "Interestingly, Affinity Capture-Mass Spectroscopy data revealed CIT-K and Cav3.3 as an interacting unit [166]." This study found that the gene CACNA1I (that codes for Cav3.3 channel) is one in several hundreds of genes that might interact with CIT-K but there are no further experiments to test this possibility.

Response: Indeed, we agree with the reviewer that further experiments are needed to confirm interaction of these kinases with TTCCs and results can be positive as well as negative. We included this section in our manuscript to provide our readers some insights into the identity of novel biomolecules that may have a possibility to modulate TTCCs and thereby regulate cellular physiology. It also provides researchers novel ideas for further research. Our headings are accordingly titled “Possible interaction…”.

We do agree that these are speculations and therefore taking into reviewer’s suggestion, we have further reduced this section (section 4). Hope this is agreeable to the reviewer. Please see Pages 12-14.

Opinion 3. Figs. 1 and 2 are too basic that they do not add anything to the review. It would be more informative and illustrative to show in Fig. 1, for instance, the potential residues/regions of T-type protein(s) that are targets for the kinases or phosphatases that are mentioned on the main text. Also Fig. 2 could be improved in some way.

Response: We thank the reviewer for this comment. We have revised old Figure 1 (Figure 2 in the revised manuscript) to contain sites on which different kinases and phosphatases act to modulate TTCCs. Since it was not possible to improve old figure 2 in a way that would be informative to our readers, we have excluded it from the revised manuscript.  

Instead, we have provided new Figure 1 (voltage-gated ion channel classification for novices) and Figure 3 (general scheme of TTCC modulation by kinases and phosphatases) that are more informative. Hope this is agreeable to be the reviewer. 

Opinion 4. Minor points:

  1. Line 38: "...thus indicating VGCC roles in non-excitable cells as well." The only non-canonical function for excitable cells that is listed is "activation of gene transcription". I suggest listing more non-excitable related functions for Ca2+ ions. 

Response: We have now listed more functions for Ca2+ ions in the manuscript. Please see Page 1.

  1. The sentence starting in line 386:  "CRF receptors are predominantly expressed in dopaminergic neurons where TTCCs are also important in the physiology of dopaminergic neurons and therefore this PKC mediated suppression of TTCCs could be of significance in diseases such as Parkinson’s where Ca2+ homeostasis is affected." The fact that Ca2+ homeostasis is affected does not imply that T-type calcium channels might be involve. There are also intracellular calcium channels (IP3 and Ryanodine, at least) that could contribute to the calcium homeostasis. The phrase is too speculative.

Response: We have now rephrased the sentence with additional reference. Please see Page 10. Reproduced below for ready reference.

___________

“CRF receptors are predominantly expressed in the dopaminergic neurons where TTCCs are also one of the important determinants of Ca2+ homeostasis and physiology. There is evidence from rat Parkinson’s model that TTCCs are involved in the pathology of Parkinson’s disease since application of TTCC blockers showed reduced motor deficits in the Parkinson’s rats. [130]. Therefore, this PKC mediated suppression of TTCCs via CRF receptor activation could be of significance in diseases such as Parkinson’s where dopaminergic neurons are affected.” ___________________

  1. Line 568: I think authors wanted to say: Cav3.1 and Cav3.2.

Response: We meant Cav3.2 and NEK4 only as this section describes possible interaction between TTCC isoform Cav3.2 and NEK4 kinase.

Overall, we have tried our best to address reviewers’ comments and we hope that the reviewers and editors now find our article suitable for publication. 

Round 2

Reviewer 2 Report

Some of the previous issues have been addressed, however still I have a few observations mainly with Tables and Figures.

Table 1, is a copy of reference 26. Which is based on only 2 previous works; one review article (Perez-Reyes, 2003) and an experimental work in pancreatic B-cells, which reports immunofluorescence data about Cav3 channels but the one for Cav3.3 is quite weak, with no further electrophysiological data to support such result. There are more original papers with more solid data like: Talley EM, et al. J Neurosci. 1999, 19(6):1895-911.

Table 2. Evidence for contribution of Cav3 channels in these cancers is limited to proliferation and, in some cases, to apoptosis evasion. And in some Refs the results are from indirect observations (for instance Ref 32). I suggest being more cautions about the role of T-type channels in cancer as this is not a review about this topic. In addition, as these refs are not further discussed in the main text, the second paragraph of Table 2 should be rephrase in this sense.   

Figure 3. The bilayer nature of the lipid membrane is missing in the scheme. Also, the cartoon of TTCC must be done again because the shown one gives a mistaken impression that Ca2+ ions permeate just through the pore of Domain I.

Author Response

Opinion 1 Table 1, is a copy of reference 26. Which is based on only 2 previous works; one review article (Perez-Reyes, 2003) and an experimental work in pancreatic B-cells, which reports immunofluorescence data about Cav3 channels but the one for Cav3.3 is quite weak, with no further electrophysiological data to support such result. There are more original papers with more solid data like: Talley EM, et al. J Neurosci. 1999, 19(6):1895-911.

Response: We have now updated the original references. Previously we had given citation of a review, but now we have included original refs. Please see revised table 1.

Opinion 2 Table 2. Evidence for contribution of Cav3 channels in these cancers is limited to proliferation and, in some cases, to apoptosis evasion. And in some Refs the results are from indirect observations (for instance Ref 32). I suggest being more cautions about the role of T-type channels in cancer as this is not a review about this topic. In addition, as these refs are not further discussed in the main text, the second paragraph of Table 2 should be rephrase in this sense.   

Response: Thank you for this comment. We have rephrased. Please see page 4.

In previous ref 32 (Dziegielewska etal) direct evidence of involvement of TTCC in ovarian cancer cell proliferation was shown by the use of TTCC blocker mibefradil and by silencing expression. However, we agree with the reviewer and therefore we have replaced the word “progression” with “proliferation”. Hope this is agreeable to the reviewer.

Opinion 3 Figure 3. The bilayer nature of the lipid membrane is missing in the scheme. Also, the cartoon of TTCC must be done again because the shown one gives a mistaken impression that Ca2+ ions permeate just through the pore of Domain I.

Response: Figure improved